# When to restart? Exploring escalating restarts on convergence

## Abstract

Learning rate scheduling plays a critical role in the optimization of deep neural networks, directly influencing convergence speed, stability, and generalization. While existing schedulers such as cosine annealing, cyclical learning rates, and warm restarts have shown promise, they often rely on fixed or periodic triggers that are agnostic to the training dynamics, such as stagnation or convergence behavior. In this work, we propose a simple yet effective strategy, which we call Stochastic Gradient Descent with Escalating Restarts (SGD-ER). It adaptively increases the learning rate upon convergence. Our method monitors training progress and triggers restarts when stagnation is detected, linearly escalating the learning rate to escape sharp local minima and explore flatter regions of the loss landscape. We evaluate SGD-ER across CIFAR-10, CIFAR-100, and TinyImageNet on a range of architectures including ResNet-18/34/50, VGG-16, and DenseNet-101. Compared to standard schedulers, SGD-ER improves test accuracy by 0.5–4.5%, demonstrating the benefit of convergence-aware escalating restarts for better local optima.

## 1   Introduction

Deep learning has transformed the landscape of artificial intelligence, driving breakthroughs in image recognition [1], natural language understanding [2], and decision-making systems [3]. While architectural innovations have played a key role in these advances, the success of deep neural networks also hinges on effective optimization strategies. Among the many factors influencing optimization, the learning rate (LR) remains one of the most critical hyperparameters. It governs the pace at which model parameters are updated during training, and its value directly affects convergence speed, stability, and generalization. A learning rate that is too high may cause divergence, while one that is too low can lead to slow convergence and suboptimal solutions.

To address this, learning rate schedulers, that adjust the LR over time, have become central to efficient training. These schedulers are particularly important in navigating the complex geometry of high-dimensional, non-convex loss landscapes, which are not only filled with saddle points, flat regions, and numerous local minima, but are also highly rugged. Adjusting the learning rate can be viewed as controlling the level of smoothness - larger rates effectively 'zoom out' and smooth over small bumps, enabling the optimizer to traverse the surface more efficiently toward better optima. Traditional schedulers typically decay the learning rate monotonically, using heuristics such as exponential or linear decay. While effective in some settings, these approaches often struggle to escape sharp minima or saddle points, and are tightly coupled to a fixed training budget.

Recent work has explored more dynamic scheduling strategies that periodically increase the learning rate to promote exploration. Cosine annealing with warm restarts [4] (SGDR) is one such method, where the learning rate follows a cosine decay and is periodically reset to a higher value. This mechanism encourages the optimizer to escape narrow minima and explore flatter regions of the loss

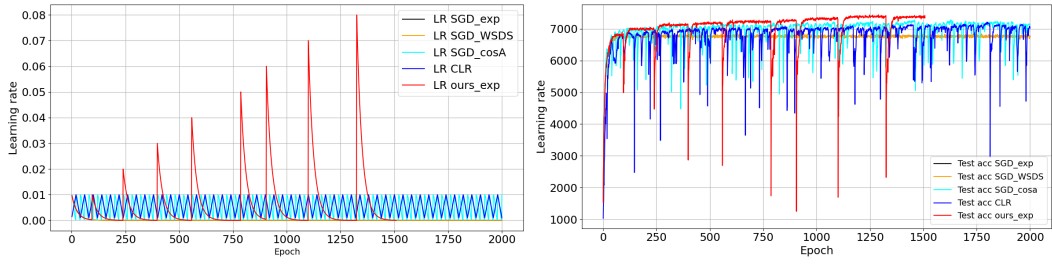

|(a) Learning rate schedulers|(b) Test accuracy on CIFAR-100 dataset.|

Figure 1: Comparison of learning rate schedulers and test accuracy on CIFAR-100 with ResNet-18 architecture with training budget of 2000 epochs. (Left) Learning rate trajectories for our proposed scheduler (ours_exp, in red) alongside four baselines: SGD with exponential decay, SGD with Warmup-Stable-Decay-Simplified (SGD_WSDS), SGD with Cyclical Learning Rate (SGD_CLR), and SGD with Cosine Annealing and Warm Restarts (SGD_cosA). (Right) Comparison of test accuracy on CIFAR-100 under identical training budgets, results highlight that our approach converges to a better local optima and terminates early when further improvements are not found.

surface. Building on this idea, cyclical learning rate (CLR) schedules proposed by Smith [5] vary the learning rate smoothly between predefined bounds without explicit restarts, effectively unifying oscillatory and restart-based behavior. More recent variants, such as Cyclical Log Annealing [6], modulate the learning rate using logarithmic functions, introducing sharper restarts and adaptive cycle amplitudes.

Warmup-Stable-Decay (WSD) scheduler [7] (Wen et al., 2024) offers a different perspective. WSD introduces a three-phase scheduler: a warm-up phase with gradually increasing LR, a stable phase with a constant high LR that allows indefinite training, and a final decay phase that sharpens convergence. This design decouples the schedule from a fixed compute budget and is motivated by a theoretical view of the loss landscape as a river valley, where progress along flat directions is made during the stable phase, and convergence in sharper directions is achieved during decay.

Despite the promise of these approaches, most current learning rate schedulers - whether cyclic, cosine-based, or with restarts rely on predefined or periodic increases in LR. These restarts are often agnostic to the actual optimization dynamics and can lead to unstable training or inefficient exploration. We argue that restarts should be adaptive: triggered by convergence rather than fixed schedules. Specifically, once the model reaches a plateau in training loss, restarting with a larger learning rate can help escape the current local minima and explore alternative regions of the loss landscape. Given the rugged, multi-modal nature of neural loss surfaces, such adaptive restarts may lead to better optima.

In this paper, we propose a simple yet effective strategy called Stochastic Gradient Descent with Escalating Restarts (SGD-ER) upon convergence. Our method detects convergence during training with some predefined patience threshold, and restarts the optimizer with a linearly increased learning rate. Each time convergence is detected, the learning rate is increased by a fixed increment, allowing the optimizer to take larger steps and escape local minima. Training continues until no further improvement is observed or a predefined budget is reached. Figure 1a shows an example of such a scheduler with exponential decay compared with its counterparts. Here, we set a patience parameter of 50, meaning that if no improvement in validation loss is observed for 50 consecutive epochs, a restart is triggered and the learning rate is linearly escalated. Figure 1b shows the comparison of test accuracies, the loss curves shows that our approach achieves better final accuracy even when other approaches were trained for 2000 epochs[1]. Figure 1b also shows that increasing the learning rate can temporarily degrade model performance; however, the model quickly recovers, and overall accuracy improves. This behavior aligns with observations in [5], where short-term instability leads to better long-term convergence.

Our experiments across standard image benchmarks show consistent accuracy gains over existing schedulers, indicating that dynamically escalating restarts can yield more effective optimization trajectories.

---

[1]The results with linear decay and Adam are given in appendix.

Table 1: Test accuracy (in %) results on CIFAR-10, CIFAR-100, and TinyImageNet with ResNet-18. Higher is better.

| Dataset | SGD_exp | SGD_lin | Adam | CosA | CLR | WSDS | Ours_exp | Ours_lin |
|---|---|---|---|---|---|---|---|---|
| CIFAR-10 | 90.86 | 91.93 | 91.34 | 92.59 | 92.15 | 93.05 | **93.83** | **93.83** |
| CIFAR-100 | 68.30 | 71.00 | 67.94 | 71.63 | 70.44 | 72.39 | **74.30** | **74.30** |
| TinyImageNet | 59.09 | 58.35 | 54.53 | 59.46 | 57.53 | 59.28 | **59.71** | **60.79** |

Table 2: Test accuracy (in %) for CIFAR-100 after 2000 epochs with ResNet-18. Higher is better.

| SGD_exp | SGD_lin | Adam | CosA | CLR | WSDS | Ours_exp | Ours_lin |
|---|---|---|---|---|---|---|---|
| 68.53 | 62.17 | 71.27 | 72.84 | 72.10 | 73.59 | **74.41** | **74.41** |

## 2 Stochastic Gradient Descent with Escalating Restarts

We introduce Stochastic Gradient Descent with Escalating Restarts (SGD-ER), a learning rate scheduler that adaptively increases the learning rate upon convergence. Unlike fixed or periodic restart schedules (e.g., cosine annealing with warm restarts, CLR), SGD-ER triggers restarts based on optimization signals, enabling controlled exploration when progress stalls and linearly escalates the learning rate across restarts to escape sharp local minima.

Let $\eta_0$ be the initial learning rate and $\mathcal{C}$ a convergence criterion (e.g., stagnation in validation loss or gradient norm). SGD-ER proceeds as follows:

- Initialize model parameters $\theta_0$ and learning rate $\eta = \eta_0$.
- Train using SGD until $\mathcal{C}$ is satisfied.
- Upon convergence, restart the optimizer with learning rate $\eta_k = (k+1) \cdot \eta_0$, where $k$ is the number of restarts.
- Continue training until either:
  - the current post-restart loss does not improve over the best loss from previous restarts, or
  - the maximum number of epochs is reached.

Each restart retains model parameters while increasing the learning rate, enabling controlled exploration of new regions in the loss landscape. The escalation is halted either when subsequent restarts fail to yield improved optima or when a predefined exploration limit is reached, thereby preventing unnecessary divergence.

**Theorem 1.** *Let $f : \mathbb{R}^d \to \mathbb{R}$ be an L-smooth function. Let $\theta^\star$ be a strict saddle point where $\nabla f(\theta^\star) = 0$ and $\lambda_{\min}(\nabla^2 f(\theta^\star)) = -\gamma < 0$. Let $\theta_0^{(k)}$ be the starting point for restart $k$. Assume there exists a non-zero residual offset $x_0 = \langle \theta_0^{(k)} - \theta^\star, v_- \rangle \neq 0$ projected onto the unstable eigenvector $v_-$.*

*During restart $k$, the SGD-ER update is:*

$$\theta_{t+1} = \theta_t - \eta_k \nabla f(\theta_t), \quad \eta_k = (k+1)\eta_0.$$

*Then, for any neighborhood radius $\delta > |x_0|$, the number of iterations $T_k$ required to escape the neighborhood $\mathcal{N}_\delta = \{\theta : \|\theta - \theta^\star\| \leq \delta\}$ satisfies:*

$$T_k \geq \frac{\ln(\delta/|x_0|)}{\ln(1 + \eta_k \gamma)},$$

*and $T_k \to 0$ as $k \to \infty$.* **Proof** *is given in Appendix B.*

## 3 Experimental Analysis

We conduct extensive experiments to evaluate the effectiveness of SGD-ER across multiple datasets and architectures. Specifically, we use CIFAR-10 [8], CIFAR-100 [8], and TinyImageNet [9], and

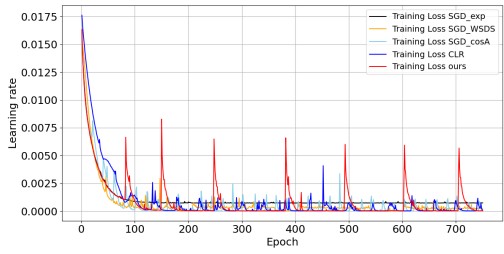

(a) Comparison of training loss.

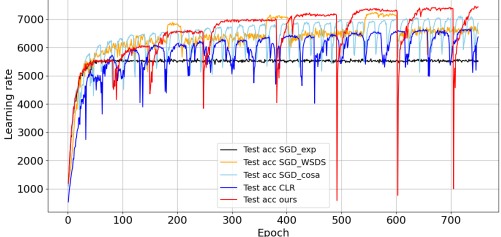

(b) Test accuracy on CIFAR-100 dataset.

Figure 2: (Left) Training loss trajectories for our proposed scheduler (ours_exp, in red) alongside four baselines: SGD with exponential decay, SGD with Warmup-Stable-Decay-Simplified (SGD_WSDS), SGD with Cyclical Learning Rate (SGD_CLR), and SGD with Cosine Annealing and Warm Restarts (SGD_cosA). (Right) Test accuracy on CIFAR-100 under a training budget of 750 epochs; our approach (in red) finds better local optima.

Table 3: Comparison of different learning rate schedulers on CIFAR-100 with ResNet-18 architecture. Values are mean ± std of best utility scores across 3 different seeds.

| Approach | Train Loss | Val Loss | Test Loss | Test Acc (%) |
|---|---|---|---|---|
| CLR | **1.60e-05 (±5.82e-07)** | 0.00488 (±6.88e-05) | 0.00496 (±3.10e-05) | 70.65 (±0.19) |
| SGD_exp | 1.11e-04 (±5.71e-06) | 0.00519 (±1.58e-04) | 0.00525 (±9.74e-05) | 68.49 (±0.17) |
| SGD_lin | 2.80e-05 (±6.74e-07) | 0.00486 (±1.03e-04) | 0.00490 (±6.29e-05) | 71.08 (±0.08) |
| Adam | 1.58e-04 (±1.22e-04) | 0.00588 (±1.44e-04) | 0.00590 (±2.02e-04) | 64.04 (±3.38) |
| CosA | 1.75e-05 (±1.63e-06) | 0.00466 (±1.79e-04) | 0.00472 (±1.27e-04) | 72.05 (±0.43) |
| WSDS | 1.64e-05 (±1.23e-06) | 0.00462 (±7.64e-05) | 0.00465 (±8.62e-05) | 72.83 (±0.44) |
| Ours_exp | 2.40e-05 (±1.19e-06) | **0.00434 (±1.74e-04)** | **0.00443 (±1.63e-04)** | **73.62 (±0.77)** |
| Ours_lin | 2.16e-05 (±1.54e-07) | **0.00427 (±1.05e-04)** | **0.00435 (±7.51e-05)** | **74.61 (±0.35)** |

test on a range of convolutional architectures including ResNet-18 [10], ResNet-34 [10], ResNet-50 [10], VGG-16 [11], and DenseNet-101 [12]. Our method is compared against several widely adopted baselines: SGD with linear decay, exponential decay, cyclical learning rate (CLR), cosine annealing (CosA), Adam, and SGD with Warmup-Stable-Decay (SGD-WSD).

Table 4: Test accuracy (in %) results on CIFAR-10 and CIFAR-100 across architectures with exponential decay.

| | CIFAR-10 | | | | | CIFAR-100 | | | | |
|---|---|---|---|---|---|---|---|---|---|---|
| Architecture | SGD_exp | CosA | CLR | WSDS | Ours | SGD_exp | CosA | CLR | WSDS | Ours |
| ResNet34 | 90.81 | 92.73 | 92.38 | 92.88 | **93.82** | 67.75 | 72.17 | 71.04 | 72.36 | **74.24** |
| ResNet50 | 90.16 | 92.77 | 91.77 | 93.13 | **94.65** | 65.52 | 72.10 | 70.25 | 73.76 | **76.77** |
| VGG16 | 90.27 | 91.42 | 90.88 | 91.44 | **92.02** | 65.17 | 67.35 | 67.23 | 68.08 | **68.56** |
| DenseNet121 | 86.31 | 91.81 | 88.99 | 92.77 | **94.19** | 56.10 | 71.20 | 66.61 | 72.45 | **76.76** |

For all SGD-based optimizers, the initial learning rate is set to 0.01, while Adam uses 0.001. The training budget is set to 750 epochs for CIFAR-100, and 500 epochs for CIFAR-10 and TinyImageNet. Additionally, we perform a long-run experiment with 2000 epochs on CIFAR-100 to assess long-term convergence behavior. All experiments are conducted using fixed random seeds for reproducibility. For CIFAR-100, we also show results averaged (along with standard deviations) over three different random seeds to prove the validy of SGD-ER across different runs.

Table 1 presents test accuracy comparisons on CIFAR-10, CIFAR-100, and TinyImageNet using ResNet-18. Our method (SGD-ER) with both exponential decay (Ours_exp) and linear decay (Ours_lin), consistently outperforms all baselines, including state of the art schedulers such as CLR, cosine annealing and WSDS. Additional utility metrics i.e., training loss, validation loss, and test loss, for these datasets are provided in the appendix.

Table 5: Test accuracy (in %) results on TinyImageNet across ResNet architectures with exponential decay.

| Dataset | ResNet-34 | | | | | ResNet-50 | | | | |
|---|---|---|---|---|---|---|---|---|---|---|
| | SGD_exp | CosA | CLR | WSDS | Ours | SGD_exp | CosA | CLR | WSDS | Ours |
| TinyImageNet | 58.08 | 61.12 | 60.30 | 62.30 | **64.53** | 58.43 | 62.50 | 60.62 | 64.52 | **64.93** |

Table 2 shows test accuracy comparisons after 2000 epochs on CIFAR-100. SGD-ER continues to outperform all baselines, demonstrating superior long-term convergence. Table 3 further compares utility metrics on CIFAR-100, showing that SGD-ER achieves lower validation and test loss with reduced variance. While CLR and CosA achieve lower training loss, they exhibit higher validation and test loss, indicating overfitting.

Table 4 reports test accuracy across architectures for CIFAR-10 and CIFAR-100 using exponential decay. Table 5 presents results on TinyImageNet with ResNet-34 and ResNet-50. In all cases, SGD-ER achieves the highest test accuracy, confirming its robustness and generalization across datasets and architectures.

## 4 Conclusion

We introduced Stochastic Gradient Descent with Escalating Restarts (SGD-ER), a simple yet effective learning rate scheduling strategy that adaptively increases the learning rate once the model reaches a plateau. Unlike fixed or periodic restart-based methods, SGD-ER leverages stagnation in the validation loss to trigger restarts and linearly escalates the learning rate to escape sharp local minima and explore flatter regions of the loss landscape. Extensive experiments across CIFAR-10, CIFAR-100, and TinyImageNet, using a variety of architectures, demonstrate consistent improvements in test accuracy ranging from 0.5% to 4.5% over its existing counterparts. These results highlight the promise of convergence-aware restarts as a lightweight mechanism for improving optimization and generalization. Future work will address the transient accuracy drops observed after learning-rate restarts by exploring smoother escalation schemes and adaptive restart thresholds.

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

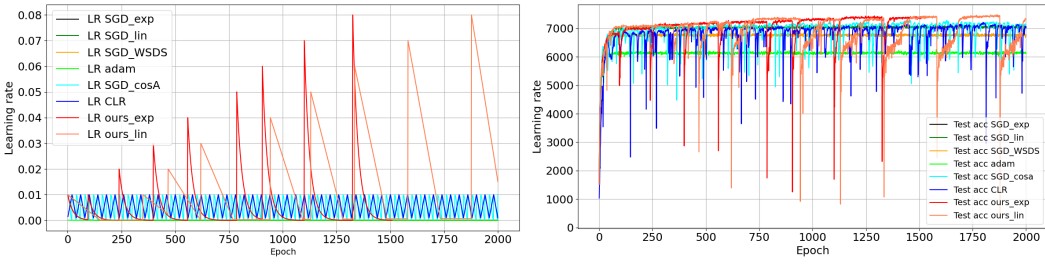

| (a) Learning rate schedulers | (b) Test accuracy on CIFAR-100 dataset. |

Figure 3: Comparison of learning rate schedulers and test accuracy on CIFAR-100 with ResNet-18 architecture with training budget of 2000 epochs. (Left) Learning rate trajectories for our proposed scheduler (ours_exp - with exponential decay, ours_lin - with linear decay) alongside these baselines: SGD with exponential decay, SGD with linear decay (SGD_lin), SGD with Warmup-Stable-Decay-Simplified (SGD_WSDS), Adam, SGD with Cyclical Learning Rate (SGD_CLR), and SGD with Cosine Annealing and Warm Restarts (SGD_cosA). (Right) Comparison of test accuracy on CIFAR-100 under identical training budgets, results highlight that our approaches converges to a better solutions.

[11] Simonyan, K. and Zisserman, A., 2015, April. Very deep convolutional networks for large-scale image recognition. In 3rd International Conference on Learning Representations (ICLR 2015). Computational and Biological Learning Society.

[12] Huang, G., Liu, Z., Van Der Maaten, L. and Weinberger, K.Q., 2017. Densely connected convolutional networks. In Proceedings of the IEEE conference on computer vision and pattern recognition (pp. 4700-4708).

## A    Additional Experimental Results

This section presents additional experimental results. Figure 3 illustrates how existing learning rate schedulers adjust the learning rate over time, while our proposed schedulers adaptively increase the learning rate linearly after each convergence is observed. In red, we show exponential decay, and in coral, linear decay after each restarts. As seen in Figure 3b, our approaches with both exponential and linear decay consistently outperform existing methods. Figure 4a depicts the learning rate curves, and Figure 4b demonstrates the improved test accuracy achieved by our approach.

Table 6 reports test, validation, and training losses for CIFAR-10, CIFAR-100, and TinyImageNet using the ResNet-18 architecture. Our approach achieves the lowest test and validation losses across all datasets. In contrast, Cyclic Learning Rate (CLR) obtains the best training losses but exhibits higher test and validation losses, indicating potential overfitting. This trend is also evident in Table 7 and Table 9, which present losses for CIFAR-100 with ResNet-18 and TinyImageNet with ResNet-34/50 respectively. Again, our methods achieve the best test and validation losses, while CLR performs better on training loss, suggesting potential overfitting.

Finally, Table 8 shows the test, validation, and training losses for CIFAR-10 and CIFAR-100 across multiple architectures (ResNet-34/50, VGG16, DenseNet121) using exponential decay. Our approaches consistently deliver the best test and validation losses, whereas CLR and WSDS often achieve lower training losses, further suggesting overfitting across diverse architectures.

Figure 5 shows an interesting budget an interesting comparison where SGD-ER approaches keeps on improving the test accuracies however the other approaches converges with the increased number of training epochs.

## B    Theoretical Analysis

**Theorem 2.** *Let $f : \mathbb{R}^d \to \mathbb{R}$ be an $L$-smooth function. Let $\theta^\star$ be a strict saddle point where $\nabla f(\theta^\star) = 0$ and $\lambda_{\min}(\nabla^2 f(\theta^\star)) = -\gamma < 0$. Let $\theta_0^{(k)}$ be the starting point for restart $k$. Assume there exists a non-zero residual offset $x_0 = \langle \theta_0^{(k)} - \theta^\star, v_- \rangle \neq 0$ projected onto the unstable eigenvector $v_-$.*

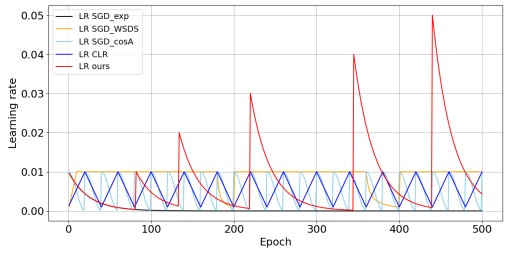
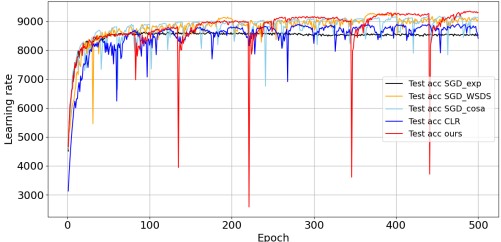

(a) Comparison of training loss.     (b) Test accuracy on CIFAR-10 dataset.

Figure 4: (Left) Training loss trajectories for our proposed scheduler (ours_exp, in red) alongside four baselines: SGD with exponential decay, SGD with Warmup-Stable-Decay-Simplified (SGD_WSDS), SGD with Cyclical Learning Rate (SGD_CLR), and SGD with Cosine Annealing and Warm Restarts (SGD_cosA). (Right) Test accuracy on CIFAR-10 under a training budget of 500 epochs; our approach (in red) finds better local optima.

Table 6: Loss values (Test, Validation, Train) on CIFAR-10, CIFAR-100, and TinyImageNet with ResNet-18. Lower is better.

| Dataset | SGD_exp | SGD_lin | Adam | CosA | CLR | WSDS | Ours_exp | Ours_lin |
|---|---|---|---|---|---|---|---|---|
| **Test Losses** | | | | | | | | |
| CIFAR-10 | 1.57E-03 | 1.51E-03 | 1.54E-03 | 1.22E-03 | 1.47E-03 | 1.19E-03 | **1.03E-03** | **1.03E-03** |
| CIFAR-100 | 5.37E-03 | 4.96E-03 | 5.67E-03 | 4.86E-03 | 4.96E-03 | 4.70E-03 | **4.29E-03** | **4.29E-03** |
| TinyImageNet | 2.65E-02 | 7.63E-03 | 9.08E-03 | 7.05E-03 | 7.73E-03 | 7.58E-03 | **7.39E-03** | **7.37E-03** |
| **Val Losses** | | | | | | | | |
| CIFAR-10 | 1.52E-03 | 1.40E-03 | 1.49E-03 | 1.22E-03 | 1.39E-03 | 1.22E-03 | **1.04E-03** | **1.04E-03** |
| CIFAR-100 | 5.37E-03 | 4.98E-03 | 5.72E-03 | 4.86E-03 | 4.95E-03 | 4.71E-03 | **4.22E-03** | **4.22E-03** |
| TinyImageNet | 2.64E-02 | 7.59E-03 | 9.02E-03 | 7.02E-03 | 7.71E-03 | 7.54E-03 | **7.38E-03** | **7.34E-03** |
| **Train Losses** | | | | | | | | |
| CIFAR-10 | 3.66E-05 | 1.19E-05 | 2.98E-05 | 1.00E-05 | **7.34E-06** | 7.83E-06 | 8.04E-06 | 8.04E-06 |
| CIFAR-100 | 1.06E-04 | 2.85E-05 | 1.75E-05 | 1.66E-05 | **1.58E-05** | 1.59E-05 | 2.50E-05 | 2.50E-05 |
| TinyImageNet | 3.19E-04 | 6.94E-05 | 2.71E-05 | 2.59E-05 | **2.54E-05** | 6.67E-05 | 6.83E-05 | 3.99E-05 |

*During restart $k$, the SGD-ER update is:*

$$\theta_{t+1} = \theta_t - \eta_k \nabla f(\theta_t), \quad \eta_k = (k+1)\eta_0.$$

*Then, for any neighborhood radius $\delta > |x_0|$, the number of iterations $T_k$ required to escape the neighborhood $\mathcal{N}_\delta = \{\theta : \|\theta - \theta^\star\| \leq \delta\}$ satisfies:*

$$T_k \geq \frac{\ln(\delta/|x_0|)}{\ln(1 + \eta_k \gamma)},$$

*and $T_k \to 0$ as $k \to \infty$.*

**Proof:**

Let $u_t = \theta_t - \theta^\star$ be the displacement from the saddle point. Since $f$ is $L$-smooth and $\nabla f(\theta^\star) = 0$, we apply a first-order Taylor expansion to the gradient:

$$\nabla f(\theta_t) = \nabla^2 f(\theta^\star) u_t + \mathcal{O}(\|u_t\|^2).$$

Substituting this into the update rule:

$$u_{t+1} = u_t - \eta_k(\nabla^2 f(\theta^\star) u_t + \mathcal{O}(\|u_t\|^2)) \approx (I - \eta_k H) u_t,$$

where $H = \nabla^2 f(\theta^\star)$. Let $v_-$ be the unit eigenvector of $H$ corresponding to the eigenvalue $-\gamma$. We project the displacement onto this direction: $x_t = \langle u_t, v_- \rangle$. The projected dynamics follow:

$$x_{t+1} = \langle (I - \eta_k H) u_t, v_- \rangle = x_t - \eta_k(-\gamma x_t) = (1 + \eta_k \gamma) x_t.$$

Table 7: Loss metrics for CIFAR-100 after 2000 epochs with ResNet-18. Lower is better.

| Loss type | SGD_exp | SGD_lin | Adam | CosA | CLR | WSDS | Ours_exp | Ours_lin |
|---|---|---|---|---|---|---|---|---|
| Test loss | 5.36E-03 | 4.96E-03 | 6.05E-03 | 4.73E-03 | 4.86E-03 | 4.58E-03 | **4.30E-03** | **4.30E-03** |
| Val loss | 5.34E-03 | 5.01E-03 | 6.14E-03 | 4.77E-03 | 4.93E-03 | 4.62E-03 | **4.29E-03** | **4.29E-03** |
| Train loss | 1.02E-04 | 2.51E-05 | 2.22E-04 | 1.56E-05 | **1.29E-05** | 1.56E-05 | 2.50E-05 | 2.50E-05 |

Table 8: Loss values (Test, Validation, Train) for CIFAR-10 and CIFAR-100 across architectures. Lower is better.

| Architecture | CIFAR-10 | | | | | CIFAR-100 | | | | |
|---|---|---|---|---|---|---|---|---|---|---|
| | SGD_exp | CosA | CLR | WSDS | Ours | SGD_exp | CosA | CLR | WSDS | Ours |
| **Test Losses** | | | | | | | | | | |
| ResNet34 | 1.70E-03 | 1.37E-03 | 1.52E-03 | 1.19E-03 | **1.09E-03** | 5.77E-03 | 4.73E-03 | 5.19E-03 | 4.91E-03 | **4.49E-03** |
| ResNet50 | 1.67E-03 | 1.34E-03 | 1.60E-03 | 1.33E-03 | **8.50E-04** | 6.26E-03 | 4.71E-03 | 5.29E-03 | 4.53E-03 | **3.97E-03** |
| VGG16 | 1.78E-03 | 1.63E-03 | 1.73E-03 | 1.64E-03 | **1.58E-03** | 6.46E-03 | 6.12E-03 | 6.70E-03 | 6.58E-03 | **6.52E-03** |
| DenseNet121 | 2.12E-03 | 1.33E-03 | 1.99E-03 | 1.27E-03 | **1.03E-03** | 6.86E-03 | 5.78E-03 | 6.67E-03 | 5.84E-03 | **3.91E-03** |
| **Validation Losses** | | | | | | | | | | |
| ResNet34 | 1.64E-03 | 1.33E-03 | 1.44E-03 | 1.22E-03 | **1.06E-03** | 5.88E-03 | 4.78E-03 | 5.16E-03 | 4.99E-03 | **4.57E-03** |
| ResNet50 | 1.58E-03 | 1.26E-03 | 1.55E-03 | 1.22E-03 | **8.80E-04** | 6.29E-03 | 4.76E-03 | 5.42E-03 | 4.56E-03 | **3.91E-03** |
| VGG16 | 1.68E-03 | 1.54E-03 | 1.63E-03 | 1.59E-03 | **1.56E-03** | 6.50E-03 | 6.05E-03 | 6.65E-03 | 6.56E-03 | **6.51E-03** |
| DenseNet121 | 2.00E-03 | 1.25E-03 | 1.84E-03 | 1.21E-03 | **1.07E-03** | 7.00E-03 | 5.76E-03 | 6.68E-03 | 5.76E-03 | **3.95E-03** |
| **Train Losses** | | | | | | | | | | |
| ResNet34 | 2.24E-05 | 1.23E-05 | **8.51E-06** | 1.01E-05 | 8.80E-06 | 5.30E-05 | 1.37E-05 | **1.06E-05** | 1.56E-05 | 1.83E-05 |
| ResNet50 | 2.35E-05 | 1.57E-05 | **1.13E-05** | 1.27E-05 | 1.42E-05 | 5.60E-05 | 1.48E-05 | **1.16E-05** | 1.61E-05 | 2.02E-05 |
| VGG16 | 3.42E-05 | 1.66E-05 | 1.80E-05 | 1.73E-05 | **1.65E-05** | 5.61E-05 | 3.11E-05 | **1.88E-05** | 2.69E-05 | 2.45E-05 |
| DenseNet121 | 7.55E-05 | 1.33E-05 | 1.07E-05 | **1.01E-05** | 1.16E-05 | 7.21E-04 | 2.10E-05 | 1.99E-05 | 2.00E-05 | **1.80E-05** |

200 Solving the linear recurrence for $t$ iterations within restart $k$:

$$x_t = (1 + \eta_k \gamma)^t x_0.$$

201 Since $\eta_k > 0$ and $\gamma > 0$, the growth factor $\alpha_k = (1 + \eta_k \gamma)$ is strictly greater than 1. This implies
202 that any non-zero initial offset $x_0$, regardless of how small, will be amplified exponentially over time.

203 Escape is defined as the first time step $T_k$ such that $|x_{T_k}| \geq \delta$. Substituting the growth equation:

$$(1 + \eta_k \gamma)^{T_k} |x_0| \geq \delta.$$

204 Taking the natural logarithm on both sides:

$$T_k \ln(1 + \eta_k \gamma) \geq \ln \left( \frac{\delta}{|x_0|} \right) \implies T_k \geq \frac{\ln(\delta/|x_0|)}{\ln(1 + \eta_k \gamma)}.$$

205 As the restart index $k$ increases, the learning rate $\eta_k = (k + 1)\eta_0$ increases linearly. Because
206 the denominator $\ln(1 + \eta_k \gamma)$ is strictly increasing with $\eta_k$, the required escape time $T_k$ decreases
207 monotonically:

$$\lim_{k \to \infty} T_k = \lim_{\eta_k \to \infty} \frac{\ln(\delta/|x_0|)}{\ln(1 + \eta_k \gamma)} = 0.$$

208 This proves that the escalation of the learning rate ensures that the optimizer will eventually escape
209 the saddle point within any predefined iteration budget $T_{max}$, provided $k$ is sufficiently large. $\qquad \square$

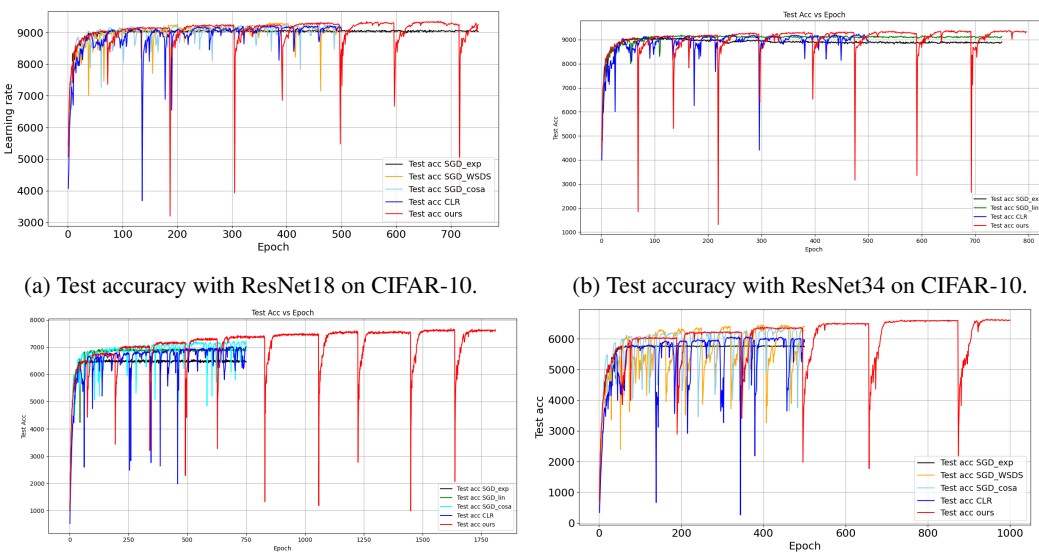

(a) Test accuracy with ResNet18 on CIFAR-10.  (b) Test accuracy with ResNet34 on CIFAR-10.

(c) Test accuracy with ResNet50 on CIFAR-100.  (d) Test accuracy with ResNet50 on TinyImageNet.

Figure 5: (Left) Training loss trajectories for our proposed scheduler (ours_exp, in red) alongside four baselines: SGD with exponential decay, SGD with Warmup-Stable-Decay-Simplified (SGD_WSDS), SGD with Cyclical Learning Rate (SGD_CLR), and SGD with Cosine Annealing and Warm Restarts (SGD_cosA). (Right) Test accuracy on CIFAR-100 under a training budget of 750 epochs; our approach (in red) finds better local optima.

Table 9: Loss values (Test, Val, and Train) on TinyImageNet across ResNet architectures. Lower is better.

| | ResNet-34 | | | | | ResNet-50 | | | | |
|---|---|---|---|---|---|---|---|---|---|---|
| Losses | SGD_exp | CosA | CLR | WSDS | Ours | SGD_exp | CosA | CLR | WSDS | Ours |
| Test loss | 8.35E-03 | 7.25E-03 | 7.35E-03 | 7.21E-03 | **2.57E-02** | 8.80E-03 | 6.62E-03 | 7.16E-03 | 6.54E-03 | **6.39E-03** |
| Val loss | 8.32E-03 | 7.16E-03 | 7.27E-03 | 7.17E-03 | **2.58E-02** | 8.80E-03 | 6.62E-03 | 7.08E-03 | 6.53E-03 | **6.35E-03** |
| Train Loss | 7.80E-05 | 1.95E-05 | **1.81E-05** | 2.10E-05 | 1.80E-04 | 6.63E-05 | 2.56E-05 | **2.15E-05** | 2.58E-05 | 4.06E-05 |