# OpenReview forum: "When to restart? Exploring escalating restarts on convergence"
_ICLR.cc/2026/Workshop/Sci4DL — Sci4DL 2026_

### Official Review · Reviewer_o1aU · 2026-02-21

**Fit:** 2
**Significance:** 2
**Confidence:** 2

**Summary:**

This paper proposes to increase the learning rate whenever stagnation is detected in the validation loss.  This method is shown to improve test accuracy.   Whenever the learning rate is increased, the model jolts out of the current 'basin' and there is an initial setback in train loss, but this ultimately leads to better generalization.

**Strengths:**

The proposed idea is interesting, and it is noteworthy that the approach works in the settings considered (image classification with different architectures).

**Suggestions:**

It would be interesting to know whether or not the method works in settings beyond image classification such as LLM pretraining or fine-tuning.  At least in LLM pretraining, there has been less evidence (relative to image classification) that larger learning rates help with generalization.

Regarding Theorem 1: I believe that the local "exploration" here is largely due to the large, positive eigenvalues, rather than the negative ones, as is argued by the theorem.

**In terms of workshop fit**: this paper is primarily about proposing an algorithm, rather than asking "scientific" questions per se.  Of course, the knowledge that a particular algorithm works is itself a very meaningful data point for science of deep learning, and for that reason the submission might potentially be relevant to this workshop.

---

### Official Review · Reviewer_keZe · 2026-02-26

**Fit:** 2
**Significance:** 2
**Confidence:** 2

**Summary:**

The paper proposes a learning rate schedule that increases the learning rate once convergence is detected. This scheduler is therefore adaptive, distinct from common schedulers, and is expected to enable better exploration of the loss landscape. The authors report improved test accuracy across various image classification benchmarks.

**Strengths:**

The proposed method is simple yet effective, and appears to deliver better performances.

**Suggestions:**

The figures have (nearly) all the y-label wrong. Fig. 4-left and Fig. 5-left don't correspond to the caption description.
The analysis considers the escape from a saddle along a negative eigenvalue direction, but for large learning rates this is usually associated with oscillations along the positive eigenvalue ones.
As the method's strength lies in its adaptivity, it would be interesting to see more comparisons with Adam and to check whether the performance gap can be closed by choosing appropriate hyperparameters.

Although the paper does not directly address the science of deep learning, the proposed method is potentially interesting in relation to the properties of the loss landscape.

---

### Official Review · Reviewer_PvMs · 2026-02-27

**Fit:** 2
**Significance:** 1
**Confidence:** 3

**Summary:**

This paper proposes SGD-ER, an LR scheduler that is designed to escape sharp local minima and find flatter regions. It monitors convergence and restarts when the stagnation criterion is satisfied, linearly increasing the LR. The method is evaluated on CIFAR-10/100 and TinyImageNet across several CNN architectures, and is reported to yield test accuracy improvements over other conventional LR schedulers including exponential/linear decay, cosine annealing, CLR, and WSD. A short theoretical argument is given to suggest that escalating the learning rate facilitates faster escape from strict saddle neighborhoods.

**Strengths:**

1. The proposed scheduler, SGD-ER, is easy to follow and intuitive. Its implementation is easy to plug into practical usage.
2. The authors support their method with a theoretical argument.

**Suggestions:**

1. **Some issues with experimental setups and results**
- The most critical issue with the experimental setup is that the initial learning rate is not swept; they only use fixed initial LRs (0.01 for SGD, 0.001 for Adam). Without sweeping the initial LRs, we cannot fairly compare the schedulers' performance. Since the paper compares SGD-ER with other LR decay schemes, a fixed initial LR would largely underestimate the performance of traditional decay schedulers. More extensive searches for optimal initial LRs would support the efficacy of the proposed method.
- In Figure 3 (b) and Figure 4 (b), they have the wrong y-axis label, "Learning rate," instead of "Test accuracy," and the values even range from 3000-9000, which is an implausible test accuracy. Also, in Table 9, the reported test and validation losses of the proposed method on ResNet-34 are far larger (2.57E-02, 2.58E-02) than those of other methods (loss<1E-02), but they are marked in bold (which typically means the lowest). Correcting these errors would support the credibility of the paper.

2. **Implementation details**: The proposed algorithm triggers restarts based on a certain "convergence criterion," but the authors do not explicitly report what kind of criterion they used in the experiments. Since the convergence criterion is the key ingredient of SGD-ER, more details should be provided.

3. **More discussion on previous works**: The idea of changing the LR while monitoring training progress has been previously investigated. I believe that the proposed LR scheduler is highly related to the well-known ReduceLROnPlateau scheduler in PyTorch. The key difference is that ReduceLROnPlateau reduces the LR, while the proposed SGD-ER escalates the LR and restarts scheduling. Comparing the proposed method with these previous schedulers would enhance the paper's impact.

4. **Connecting to sharpness dynamics**: The choice of LR and its optimization dynamics on the loss landscape have been investigated in various works. Recent perspectives suggest that choosing a large LR guides optimization trajectories to flat regions, staying near the edge of stability [1]. To understand the effectiveness of SGD-ER, it would be interesting to track the sharpness dynamics when using SGD-ER and its relationship with the model's generalization capability.

[1] Cohen et al., Gradient descent on neural networks typically occurs at the edge of stability, ICLR 2021.

---

### Meta-Review · Area_Chair_Wb2t · 2026-03-02

**Recommendation:** Accept

**Metareview:**

The paper proposes a learning rate schedule that increases the learning rate once convergence is detected. The method is sound and can be of interest for the workshop.

---

### Decision · Program_Chairs · 2026-03-02

Accept